# Acute Kidney Injury is Associated with Lowered Plasma-Free Thiol Levels

**DOI:** 10.3390/antiox9111135

**Published:** 2020-11-16

**Authors:** Lisanne Boekhoud, Jacqueline Koeze, Elisabeth C. van der Slikke, Arno R. Bourgonje, Jill Moser, Jan G. Zijlstra, Anneke C. Muller Kobold, Marian L. C. Bulthuis, Matijs van Meurs, Harry van Goor, Hjalmar R. Bouma

**Affiliations:** 1Department of Clinical Pharmacy and Pharmacology, University of Groningen, University Medical Center Groningen, 9713GZ Groningen, The Netherlands; l.boekhoud@umcg.nl (L.B.); e.c.van.der.slikke@umcg.nl (E.C.v.d.S.); 2Department of Critical Care, University of Groningen, University Medical Center Groningen, 9713GZ Groningen, The Netherlands; j.koeze@umcg.nl (J.K.); j.moser@umcg.nl (J.M.); j.g.zijlstra@umcg.nl (J.G.Z.); m.van.meurs@umcg.nl (M.v.M.); 3Department of Gastroenterology and Hepatology, University of Groningen, University Medical Center Groningen, 9713GZ Groningen, The Netherlands; a.r.bourgonje@umcg.nl; 4Department of Pathology and Medical Biology, University of Groningen, University Medical Center Groningen, 9713GZ Groningen, The Netherlands; m.bulthuis01@umcg.nl (M.L.C.B.); h.van.goor@umcg.nl (H.v.G.); 5Department of Laboratory Medicine, University of Groningen, University Medical Center Groningen, 9713GZ Groningen, The Netherlands; a.c.muller@umcg.nl; 6Department of Internal Medicine, University of Groningen, University Medical Center Groningen, 9713GZ Groningen, The Netherlands

**Keywords:** sepsis, acute kidney injury (AKI), reactive oxygen species (ROS), oxidative stress, free thiols, hydrogen sulphide

## Abstract

Acute kidney injury (AKI) is associated with the abrupt loss of kidney function. Oxidative stress plays an important role in the pathophysiology of AKI. Free thiols (R-SH) are crucial components of the extracellular antioxidant machinery and reliably reflect systemic oxidative stress. Lower levels of thiols represent higher levels of oxidative stress. In this preliminary study, we hypothesized that plasma-free thiols are associated with AKI upon admission to the intensive care unit (ICU). In this study, 301 critically ill patients were included. Plasma samples were taken upon admission, and albumin-adjusted plasma-free thiols were determined. Albumin-adjusted plasma-free thiols were lower in patients with AKI (n = 43, median (interquartile range) 7.28 µmol/g (3.52, 8.95)) compared to patients without AKI (8.50 μmol/g (5.82, 11.28); *p* < 0.05) upon admission to the ICU. Higher age (B = −0.72), higher levels of neutrophil gelatinase-associated lipocalin (B = −0.002), creatinine (B = −0.01) and lower serum albumin (B = 0.47) were associated with lower free thiol levels. Further, albumin-adjusted free thiol levels were significantly reduced in patients with sepsis (8.30 (5.52–10.64) µmol/g) compared to patients without sepsis (6.95 (3.72–8.92) µmol/g; *p* < 0.05). Together, albumin-adjusted plasma-free thiols were significantly reduced in patients with AKI and patients with sepsis compared with patients without AKI and sepsis.

## 1. Introduction

Acute kidney injury (AKI) is associated with increased morbidity and mortality, both at the short and long-term [1]. AKI is defined according to the Kidney Disease: Improving Global Outcomes (KDIGO) guideline as an acute increase in serum creatinine and/or a decrease in urine output [2]. AKI occurs in approximately 10–15% of hospitalized patients, while its incidence in intensive care units (ICUs) has been reported to exceed 50% [1]. Although AKI is usually not the primary reason for ICU admission, it often complicates the clinical course of critically ill patients and is associated with an increased risk of developing end-stage renal disease (ESRD) and mortality during and after hospitalization [3]. In clinical practice, there is a lack of standardized preventive measures against AKI in critically ill patients [3]. The most common cause of AKI in critically ill patients is sepsis [4], which is a life-threatening dysregulated host response to infection, leading to organ dysfunction. Sepsis has a relatively high incidence, with 48.9 million cases every year worldwide [5], and is associated with a poor prognosis. As such, one in three patients with sepsis will decease during their hospital stay, making it the leading cause of death among patients admitted to the ICU [6]. The pathophysiology of sepsis-associated AKI consists of multiple factors, such as decreased renal blood flow, increased renal vascular resistance, endothelial dysfunction, infiltration of inflammatory cells in the renal parenchyma, and obstruction of tubules with necrotic cells [7,8]. AKI is characterized by a complex pathophysiology in which, amongst others, oxidative stress plays an important role [9].

Oxidative stress is defined as an imbalance between the production of reactive oxygen species (ROS) and a decreased availability of antioxidants [3]. Sepsis is associated with an increased production of ROS [10]. Activated immune system components and dysfunctional mitochondria play an important role in the generation of ROS in sepsis [11,12]. The overproduction of ROS leads to oxidative damage to the mitochondria, DNA, lipids, and enzymes in the renal parenchyma [10], which, together, compromise renal function [3]. Counteracting oxidative stress could be a potential strategy to prevent or treat AKI [5]. In an ICU population, substitution of the recommended daily allowance of antioxidants such as vitamin C improved the antioxidant capacity [13], and patients with sepsis who received antioxidants had a lower risk of developing AKI [14].

Measuring the critical components of redox signaling could be a possible strategy to identify patients with higher levels of oxidative stress. Thiols are central components of the extracellular nonenzymatic antioxidant machinery [15]. A thiol (R-SH, sulfhydryl group) is an organosulfur compound that can scavenge free radicals [15]. A reduction in free thiol groups reflects systemic oxidative stress, since they are prime substrates for reactive species. Thiols are a robust and powerful biomarker for an individual’s systemic reduction-oxidation (redox) status and are representative of the degree of systemic oxidative stress [16]. Other antioxidant compounds such as glutathione, homocysteine, or cysteine comprise low-molecular-weight (LMW) thiols and are of minor importance to the antioxidant capacity, as the greatest share is represented by free thiols (60–75%), and thiols are the major determinants of the total antioxidant capacity [16]. High levels of systemic free thiols, as potent antioxidant substances, are reflective of a more favorable in vivo redox status [17]. Plasma proteins, mainly albumin, contain the largest amount of redox-active thiol groups (approximately 75% of the total thiol pool) [18].

Since oxidative stress plays an important role in the etiology of AKI in critically ill patients, thiols could potentially represent a biomarker to identify patients at-risk of developing or progressing in AKI. In contrast to previous studies that primarily focused on free thiols as a disease biomarker, we will focus on thiols as a pathophysiological indicator. In this study, we therefore investigated whether thiols are associated with AKI in patients admitted to the ICU with and without sepsis. Based on the hypothesis that inflammation in sepsis augments oxidative stress [19], we also aimed to investigate associations between plasma-free thiol levels and inflammatory biomarkers like C-reactive protein (CRP), calprotectin, and neutrophil gelatinase-associated lipocalin (NGAL) [11].

## 2. Materials and Methods

### 2.1. Patient Population, Data Collection, and Definitions

Patients who were admitted to the intensive care unit (ICU) at the University Medical Center Groningen (UMCG, Groningen, Netherlands) between January 2014 and April 2014 were included in this case-control study. Clinical data and blood samples were collected daily upon admission to the ICU until discharge, or when patients had a prolonged stay, data was collected until day eight. All consecutive patients admitted to the ICU during the study period were included. If patients were admitted multiple times during the study period, only data from the first admission was used for the analysis. The Medical Ethics Review Committee (in Dutch: “Medische Ethische Toetsingscommissie” (METc)) from the UMCG reviewed and waived this study (METC 2013/174). Initially, 361 patients were included in the database. Patients of whom plasma samples were not available at day one of ICU admission were excluded (n = 60). Thiols were measured in samples derived on the first day of ICU admission, while creatinine was measured on the first and third day of ICU admission to determine AKI progression. Patients with chronic kidney disease (CKD, defined by a previously known SCr > 177 mmol/L) and patients on chronic renal replacement therapy (RRT) were excluded from the study, as were renal transplant recipients (n = 15)**.** Patients were stratified into groups based on the presence of AKI upon admission, defined by the KDIGO AKI score (Appendix A
Table A1) [2] based on the change in serum creatinine upon admission to the ICU as compared to the pre-existent value in the year preceding ICU admission. In case no pre-existent creatinine value was available, the baseline creatinine was estimated using the Modification of Diet in Renal Disease (MDRD)-based estimation method, assuming a creatinine clearance of 75 mL/min/1.73 m^2^ [20]. Besides AKI stage, continuous change in serum creatinine was used and calculated as the change in pre-existent serum creatinine and serum creatinine upon admission. After the first day of admission, when diuresis was known, AKI was defined based on the KDIGO criteria using both serum creatinine and diuresis (Appendix A
Table A1). AKI progression was defined as any increase in KDIGO stage within 48 h. Presence of sepsis was determined based on the sequential organ failure assessment (SOFA) score [21]. Sepsis was defined as two or more points in the SOFA criteria and microbiological evidence of an infection.

### 2.2. Measurement of Plasma-Free Thiol Levels

Plasma-free thiol concentrations were measured as previously described by Ellman et al., with minor modifications [22,23]. Plasma samples were stored at −80 °C until analysis of free thiol levels. First, samples were thawed on ice overnight, followed by centrifugation at 10,000 rpm for 10 min at 4 °C. A calibration curve with L-cysteine (Fluka Biochemika, Buchs, Switzerland) standard curve (15.625 µM to 1000 µM) was made in 0.1-M Tris/EDTA buffer (pH 8.2). Next, 75-µL plasma was 4-fold diluted with 0.1-M Tris/EDTA buffer (pH 8.2) and added to a flat-bottom 96-well plate in triplicates. After 20 min of incubation at room temperature, the absorbance was measured at 630 nm, while absorbance at 412 nm was subtracted as background. Next, 20 µL of 1.-mM DTNB (5,5′-dithio-bis (2-nitrobenzoic acid) Ellman’s Reagent, Sigma Aldrich Corporation, St. Louis, MO, USA) in phosphate buffer (0.1 M, pH 7) was added, followed by incubation for 20 min at room temperature in complete darkness. Again, absorbance was measured at 630 nm (reference) and 412 nm (background absorption). The concentration of free thiol levels was calculated in comparison with the calibration curve. Plasma-free thiol concentrations were corrected for plasma albumin by dividing free thiols through albumin concentrations, since albumin is the most abundant human plasma protein and the predominant source of thiols [18].

### 2.3. Measurement of Calprotectin

Serum calprotectin levels were quantified using MRP8/14 ELISA kit (BÜHLMANN, Schönenbuch, Switzerland) using the DS2 ELISA robot (DS2, Dynex, Chantilly, VA, USA) according to the manufacturer’s instructions to determine the neutrophil activation; however, they are also released though monocytes, macrophages, and squamous epithelial cells. Inter-assay coefficients of variation (CV) were 12% and 6.8% at levels of 1.47 and 5.81 ug/mL, respectively.

### 2.4. Measurement of Plasma NGAL

NGAL, a kidney injury marker, was measured in routinely collected lithium heparin plasma samples using the BioPorto NGAL Test (BioPorto Diagnostics, Hellerup, Denmark) in the Department of Laboratory Medicine on a Roche Modular P800 chemistry platform (Roche, Mannheim, Germany). According to the manufacturer, the NGAL test was validated for NGAL levels between 25 and 5000 mg/L. Overall, the CV was 2.9% at a level of 206 mg/L and 2.3% at a level of 511 mg/L.

### 2.5. Statistical Analysis

Data analysis and data visualization were performed using R studio (RStudio Team (2015). RStudio: Integrated Development for R. RStudio, Inc., Boston, MA, USA). Descriptive statistics are presented as means ± standard deviations (SD), medians (interquartile ranges, IQR) (in case of skewed distributions), and proportions n with corresponding percentages (n, %). Comparisons between groups for continuous variables were performed using independent sample *t*-tests, Mann-Whitney U tests, one-way analysis of variance (ANOVA), or Kruskal-Wallis tests, while, for nominal variables, chi-square tests or Fisher’s exact tests were performed, as appropriate. The normality testing was performed using Q-Q plots. Correlations were tested by either Pearson’s correlation coefficients (r) or Spearman’s rank correlation coefficients (ρ), depending on the normality of distribution, respectively. Associations between covariates and levels of thiols and renal function were calculated using a univariable linear regression analysis where covariates were selected, followed by a multivariable linear regression analysis of selected covariates (univariate F < 0.2). Finally, a logistic regression analysis (odds ratios with 95% confidence intervals (CI)) was performed to study the associations between covariates and the presence of AKI. Statistical significance was defined as a two-tailed *p*-value ≤ 0.05.

## 3. Results

### 3.1. Baseline Characteristics of the Study Population

Initially, a total of 361 subjects consented to participate in the study. However, 60 patients were excluded, because limited plasma samples were available. Further, 15 patients were excluded because of chronic kidney disease. Therefore, data from 286 patients were included for analysis—of which, 109 patients were female (38%; Table 1). The median age of the patients was 62 (IQR 53–71) years. In total, 33 (11%) patients had AKI upon ICU admission, with no difference in median levels of the baseline serum creatinine (83 (IQR 71–104) µmol/L) compared to 253 (88.4%) patients without AKI upon admission (78 (68–94) µmol/L; *p* > 0.05). The median creatinine levels upon ICU admission (162 (136–208) µmol/L) were higher compared to patients without AKI upon admission (69 (58–82) µmol/L; *p* < 0.05; Table 1). Diabetes mellitus (DM) was more commonly observed in patients with AKI upon admission (*p* < 0.05) compared to non-AKI patients. Of all patients, 49 (17%) were admitted with a confirmed infection, of which 40 (18%) patients were admitted with the diagnosis of sepsis. Of all patients with AKI, 15 (45%) had an infection, of which 87% were eventually classified as sepsis. In total, 44.8% of the patients with AKI had sepsis. The diagnosis of sepsis was based on two or more SOFA points, which is based on the PaO2/FiO2 ratio; thrombocyte count; bilirubin levels; mean arterial pressure (MAP); and need for vasopressors, Glasgow coma scale, and kidney function (i.e., based on serum creatinine and diuresis). Admission to the ICU after scheduled or unscheduled surgery was the case in 27% of the patients with AKI, while 72% of the patients without AKI had a postoperative admission (*p* < 0.01). Patients with AKI upon admission had higher plasma levels of C-reactive protein (CRP) and higher urinary albumin excretion (*p* < 0.001) at ICU admission. Seventeen patients (6%) died within seven days and 34 (10.5%) within 28 days of follow-up; the mortality rate at day seven and 28 was higher in patients with AKI upon admission as compared to patients without AKI upon admission (*p* < 0.05).

### 3.2. Reduced Plasma-Free Thiol Levels were Associated with AKI upon Admission

Patients admitted to the ICU with AKI had significantly lower unadjusted plasma-free thiol levels (182.75 (75.78–260.17) µmol/L) as compared to patients without AKI upon admission (275.57 (166.96–395.21) µmol/L; *p* < 0.01; Figure 1A). Similarly, after adjustment to the albumin levels, patients with AKI upon admission had lower albumin-adjusted plasma-free thiols levels (7.3 (3.5–9.0) µmol/g) compared to critically ill patients without AKI upon admission (8.5 (5.8–11.3) µmol/g; *p* < 0.05; Figure 1B). Albumin-adjusted plasma-free thiol levels inversely correlated with the change in serum creatinine from the baseline upon admission to the ICU (R = −0.20, *p* < 0.001; Figure 1C). To assess whether plasma-free thiol levels were also associated with the severity of AKI, we stratified patients into a group of severe AKI (oliguria, anuria, or need of renal replacement therapy, n = 12); nonsevere AKI (n = 21); and no AKI (n = 286) and compared plasma-free thiol levels between groups. Levels of free thiols in plasma were not different between groups (*p* > 0.05), which might, however, be due to the relatively small group size after stratification. Reduced albumin-adjusted plasma-free thiol levels (OR (odds ratio) = 0.87), increased Acute Physiology And Chronic Health Evaluation (APACHE) IV (OR = 1.03), and diabetes mellitus (OR = 2.68) were independently associated with an increased risk of AKI (Table 2). Admission to the ICU other than for scheduled or unscheduled surgery was associated with a decreased risk of AKI (OR = 0.34; Table 2). Gender, age, CRP, and sepsis were not associated with AKI in the multivariable logistic regression model.

### 3.3. Association of Plasma-Free Thiol Levels with the Course of AKI during ICU Admission

To assess whether plasma-free thiol levels could predict the course of AKI, we correlated plasma-free thiol levels measured upon admission with the serum creatinine levels that were measured daily during admission. The median serum creatinine levels upon admission were higher in patients with AKI progression (91 (72–131) µmol/L) as compared to patients without AKI progression (72 (59–88) µmol/L; *p* < 0.05; Appendix B
Table A2). In both AKI progression and new-onset AKI, no significant differences were found in albumin-adjusted plasma-free thiol levels (*p* > 0.05; Figure 1D,E).

### 3.4. Association of Plasma-Free Thiol Levels with Sepsis

Albumin-adjusted plasma-free thiol levels were significantly reduced in patients with sepsis (6.95 (3.72–8.92) µmol/g) compared to patients without sepsis (8.30 (5.52–10.64) µmol/g; *p* < 0.05; Figure 2A). When separating AKI patients into groups with and without sepsis, we observed that patients with sepsis-associated AKI had lower levels of albumin-adjusted free thiol levels (6.8 (2.0, 7.9) µmol/g) compared to patients without both sepsis and AKI (8.4 (5.8, 10.8) µmol/g; *p* < 0.05). However, we found no difference in plasma albumin-adjusted free thiol levels between patients with AKI with sepsis compared to patients with AKI without sepsis (7.0 (3.5, 9.5) µmol/g) compared to patients without AKI and/or sepsis (*p* > 0.05; Figure 2B). Together, these data suggest that free thiol levels are mainly reduced in sepsis-associated AKI.

### 3.5. Serum Calprotectin is Associated with AKI and Sepsis

Serum calprotectin was significantly increased in patients with AKI (5.06 (3.94–9.92) µg/mL) compared to patients without AKI (3.91 (2.36–6.62) µg/mL; *p* < 0.05; Figure 3A). Serum calprotectin was also significantly increased in patients with sepsis (5.14 (3.94–11.24) µg/mL) compared to patients without sepsis (3.92 (2.33–7.05) µg/mL; *p* < 0.05; Figure 3B). Together, these data suggest that calprotectin is increased during AKI and sepsis.

### 3.6. Plasma-Free Thiol Levels were Associated with Age, Serum Albumin, Creatinine Levels, and Inflammatory Parameters

Higher age (B = −0.72, *p* < 0.001), higher levels of neutrophil gelatinase-associated lipocalin (NGAL) (B = −0.002, *p* < 0.05), higher creatinine, higher CRP (B = −0.01 and B = −0.01, *p* < 0.05), and lower serum albumin (B = 0.47, *p* < 0.001; Table 3) were associated with lower plasma-free thiol levels. In contrast, plasma levels of calprotectin did not associate with plasma-free thiol levels (*p* > 0.05; Figure 4B). Of note, calprotectin and NGAL were positively correlated with each other (R = 0.23, *p* < 0.05; Figure 4C).

## 4. Discussion

In this study, we investigated the association between albumin-adjusted plasma-free thiol levels in relation to (sepsis-associated) AKI in critically ill patients as a biomarker for oxidative stress. Most importantly, we observed that patients admitted to the ICU with AKI had significantly lower levels of plasma-free thiols as compared to patients without AKI. However, plasma-free thiol levels upon admission were not significantly different between patients with and without new-onset AKI or the progression of AKI within 48 h. Furthermore, patients with sepsis had significantly reduced levels of plasma-free thiols upon admission compared to patients without sepsis. Patients with sepsis-associated AKI had lower levels of albumin-adjusted free thiol levels compared to patients without sepsis. Additionally, we observed that plasma-free thiol levels were associated with age, CRP, serum albumin, serum creatinine, and serum NGAL. In contrast, calprotectin did not correlate with plasma-free thiol levels.

Oxidative stress plays an important role in the pathogenesis of AKI in critically ill patients [3]. The excessive production of free radicals overpowering the antioxidant machinery results in oxidative stress, which is, in turn, responsible for extensive cellular and molecular damage. Furthermore, AKI itself is a stimulus for increased oxidative stress, due to mitochondrial dysfunction [3,24]. Over the past decade, multiple biomarkers for AKI have been studied and proposed [25,26]. However, no adequate and clinically applicable biomarker for the early prediction of AKI development in critically ill patients is currently available. In clinical practice, creatinine levels are nowadays used as a biomarker to diagnose AKI, but these levels only change once renal failure occurs [27]. In contrast, several other AKI biomarkers have been proposed, such as NGAL, interleukin 18 (IL-18), and urine calprotectin [25,26]. NGAL is a biomarker with a high predictive and diagnostic value for AKI, which is released during ischemia. However, it is also released during systemic inflammation and, therefore, lacks specificity [28,29]. Given the fact that oxidative stress is a key player in the etiology of AKI in critical illness, extracellular free thiol levels may be of potential diagnostic and/or predictive value [15]. In our study, we demonstrated that patients with AKI upon admission had significantly reduced levels of albumin-adjusted plasma-free thiol levels compared to patients without AKI upon admission. In addition, plasma-free thiol levels significantly correlated with indicators of renal function, including the change in creatinine upon admission. Previous research demonstrated that patients with hospital-acquired AKI had lower levels of unadjusted plasma-free thiols as compared to critically ill patients without AKI and healthy subjects [30]. In another study, pediatric patients with AKI had lower levels of unadjusted plasma-free thiols as compared to healthy controls [31]. Both the unadjusted thiol levels correlated with the albumin-adjusted thiol levels in our study; in contrast, we used adjusted thiol levels to correct for the albumin levels.

NGAL and calprotectin are associated with neutrophil activation [11,28]. Calprotectin is a protein heterodimer derived from neutrophils and monocytes, which both play a key role during inflammation by inducing the activation of immune cells and enhancing their ROS production, thereby augmenting oxidative stress [11,32]. Patients who developed AKI after cardiac surgery had higher levels of plasma calprotectin as compared to patients who did not develop AKI [33]. In the present study, we found no correlation between serum calprotectin and plasma-free thiol levels. Given the association between calprotectin with inflammation and oxidative stress, the absence of the correlation between calprotectin and free thiols in the present study was against our expectations. This could potentially be explained by the fact that neutrophil activation is of lesser importance in the case of AKI-associated oxidative stress. NGAL is another biomarker of systemic inflammation that is released from neutrophil granules. Both NGAL and calprotectin were positively correlated with each other. However, NGAL is also rapidly induced and released from injured kidney tissue and is, therefore, a less-specific biomarker of neutrophil activation [3,25,28]. Further, NGAK is an important contributor to free-radical generation [3]. As described earlier, higher levels of NGAL are associated with AKI [25,26]. Our results showed higher plasma levels of NGAL in patients with sepsis-associated AKI compared to controls. Further, in our study, plasma levels of NGAL correlated with plasma-free thiols. Together, plasma-free thiols are associated with NGAL but not with calprotectin levels. This may implicate that a decrease in plasma-free thiol levels would be more associated with a different source of free radicals than neutrophil activation.

The lowered plasma-free thiol levels in patients with AKI might either indicate the scavenging of antioxidants due to the increased oxidative stress or, vice versa, suggest a protective effect of increased plasma-free thiol levels against AKI. Yet, the current data does not allow to draw a firm conclusion about the precise role in the pathophysiology of AKI. Since plasma-free thiols were decreased in AKI, it might also identify patients at-risk for the development of AKI. However, we did not detect a significant difference in albumin-adjusted plasma-free thiol levels in patients that developed AKI compared to patients who did not develop AKI during their ICU stay. Furthermore, albumin-adjusted plasma-free thiol levels showed no significant difference between patients with AKI progression and without AKI progression during ICU admission. Based on these findings, it may be questioned if albumin-adjusted plasma-free thiol levels could be used as an early indicator of AKI development, as was originally hypothesized. This suggests that AKI itself is associated with a decrease in free thiols.

The strengths of this study comprise the large and extensively characterized study population consisting of patients who were admitted to the ICU in our university medical center. There was no selection bias, since all patients admitted to the ICU as subsequent admissions were included between January 2014 and April 2014. All relevant demographic, clinical, and biochemical information was available for each patient from admission upon to eight days. To date, to the best of our knowledge, no other study unravelled the association between (albumin-adjusted) plasma-free thiols and AKI in patients in the ICU with and without sepsis. Previous studies demonstrated lowered plasma-free thiol levels in children with AKI [31] or adult patients with AKI [30], although not among adult patients in the ICU. Moreover, as thiols are influenced by serum albumin levels, which might be affected by critical illness or sepsis, we corrected the thiol levels for albumin, in contrast to previous studies. However, our study also had several limitations that have to be considered. For instance, we did not have plasma samples available from every patient in our cohort, which downsized our sample size for the plasma-free thiol levels analyses to 83.4% of the included patients. Further, most patients stayed for a short amount of time at the ICU; therefore, we had less patients after day one in this study. Since our study was performed in a tertiary academic care center, patients enrolled in our study more frequently had complicated pathologies, including complex oncological disease and post-transplantation complications. This may limit the generalizability of our results to other studies on critically ill patients.

## 5. Conclusions

In this study, albumin-adjusted plasma-free thiol levels were significantly reduced in patients with AKI upon admission as compared to patients without AKI. However, reduced plasma-free thiol levels were not associated with the progression and course of AKI in a large cohort of critically ill patients. As thiols are central components of the extracellular antioxidant network and represent key transducing elements in redox signaling, they are indicative of oxidative stress Here, we demonstrated that plasma-free thiol levels are associated with AKI. The pathophysiology of AKI in critical illness, whether due to sepsis or other conditions, remains complex and incompletely understood. Although, we revealed a correlation between lowered plasma-free thiol levels and AKI, which suggested the hypothesis that oxidative stress plays a key role in the pathophysiology of AKI in the ICU. Although, at least for some individuals in our cohort, the lowered plasma thiol levels might well contribute to the development of AKI, the discriminative effect of plasma-free thiol levels is limited, illustrating the complex pathophysiology in this relatively heterogeneous population.

## Figures and Tables

**Figure 1 antioxidants-09-01135-f001:**
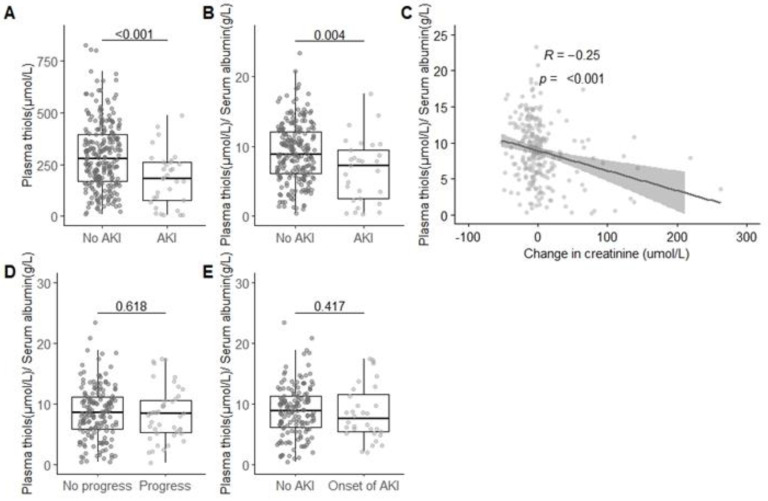
Acute kidney injury is associated with lowered plasma-free thiol levels. (**A**) Plasma-free thiol levels are lower in patients with AKI as compared to non-AKI. (**B**) After adjusting plasma-free thiol levels for the plasma albumin levels, plasma thiol levels remain lower in patients with AKI as compared to non-AKI patients. (**C**) Adjusted plasma-free thiol levels inversely correlate with a change in serum creatinine from the baseline to admission. (**D**) Adjusted plasma-free thiol levels for albumin are not different in patients with AKI progression within 48 h as compared to patients without AKI progression. (**E**) Adjusted plasma-free thiol levels for albumin are not different in patients with AKI onset within 48 h as compared to patients without AKI onset or AKI. Group differences were calculated using a two-tailed Mann-Whitney U test; correlations were calculated with Spearman.

**Figure 2 antioxidants-09-01135-f002:**
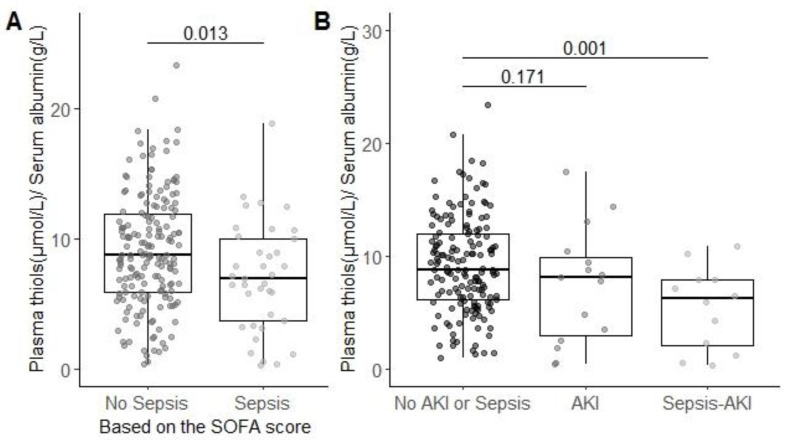
Patients with sepsis have lower albumin-adjusted free thiol levels in plasma. SOFA: sequential organ failure assessment. (**A**) Adjusting the plasma-free thiol levels for plasma albumin. Plasma thiol levels are lower in patients with sepsis as compared to patient without sepsis. (**B**) Adjusted plasma-free thiol levels are only significant lower in patients with sepsis-induced AKI compared to patients without sepsis or AKI. Group differences are calculated using a two-tailed Mann-Whitney U test.

**Figure 3 antioxidants-09-01135-f003:**
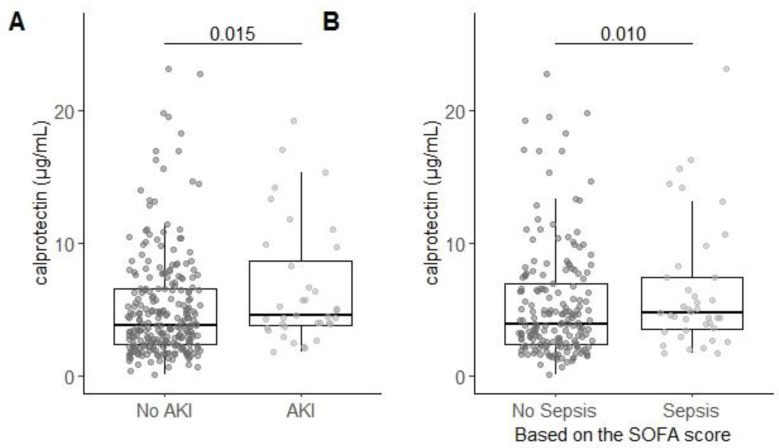
Serum calprotectin levels are correlated with AKI and sepsis. (**A**) Serum calprotectin levels were higher in patients with AKI as compared to patient without AKI. (**B**) Serum calprotectin levels were higher in patients with sepsis as compared to patient without sepsis. Group differences are calculated using a two-tailed Mann-Whitney U test.

**Figure 4 antioxidants-09-01135-f004:**
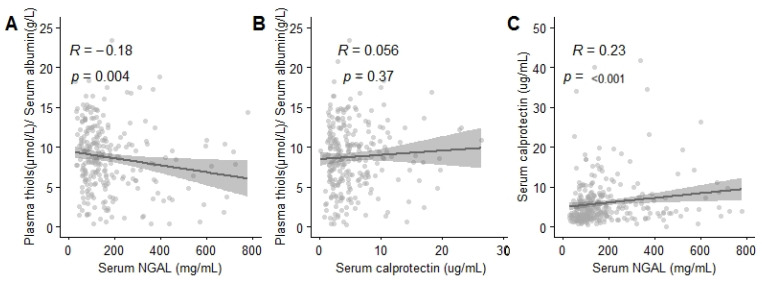
Albumin-adjusted plasma thiol levels correlate with plasma neutrophil gelatinase-associated lipocalin (NGAL). (**A**) Plasma-free thiol levels correlate with plasma NGAL levels. (**B**) Plasma-free thiol levels do not correlate with plasma calprotectin levels. (**C**) Serum calprotectin levels correlate with plasma NGAL levels. Correlations are calculated with Spearman.

**Table 1 antioxidants-09-01135-t001:** Baseline characteristics.

Characteristics	Total (*n* = 286)	No AKI at Admission (*n* = 253 (88.4%))	AKI at Admission (*n* = 33 (11.6%))	*p*-Value
Age (years)	62 (53, 71)	62 (53,70)	66 (55, 74)	0.096
Gender (% female)	109 (38.1)	96 (37.9)	13 (39.4)	1.000
BMI (kg/m^2^)	22.8 (20.7, 25,3)	22.7 (20.8, 25.0)	23.2 (20.1, 25.8)	0.664
Baseline creatinine (µmol/L)	78 (68, 95)	78 (68, 94)	83 (71, 104)	0.072
Comorbidities				
COPD (%)	34 (12.0)	28 (11.2)	9 (21.4)	0.339
CVD (%)	31 (12.0)	28 (11.2)	6 (18.8)	0.231
DM (%)	46 (7.8)	34 (13.4)	16 (37.2)	**0.002**
Malignancy (%)	22 (7.8)	21 (8.4)	1 (3.1)	0.255
Operation (%)	192 (67.1)	183 (72.3)	9 (27.3)	**<0.001**
Infection (%)	49 (17.1)	34 (13.4)	15 (45.5)	**<0.001**
Sepsis (%)	40 (18.3)	27 (14.2)	13 (44.8)	**<0.001**
SIRS score	3 (1, 3)	2 (1, 3)	3 (3, 3)	0.619
SOFA score	6 (3, 7)	5 (3, 7)	8 (6, 10)	**<0.001**
Mechanical ventilation (%)	203 (71.7)	174 (69.3)	29 (90.6)	**0.021**
RRT (within 24 h)	6 (2.1)	2 (0.8)	4 (12.1)	**<0.001**
Vital parameters				
Heart rate (bpm)	80 (70, 100)	80 (70, 100)	90 (76, 102)	0.078
MAP (mmHg)	84.0 (72.3, 98.3)	84.5 (73.3, 98.7)	78.3 (68.2, 93.8)	0.416
Respiratory rate (resp/min)	18 (15, 24)	16 (14, 20)	24 (18, 33)	**0.006**
Body temperature (°C)	36.2 (35.4, 37.3)	36.4 (35.5, 37.3)	35.4 (3501, 36.5)	0.285
Glasgow coma scale				**0.013**
15	229 (80.9)	209 (83.3)	20 (62.5)	
13–14	14 (4.9)	10 (4.0)	4 (12.5)	
≤12	40 (14.1)	32 (12.7)	8 (25.0)	
Laboratory values				
CRP (mg/L)	10.20 (2.70, 61.47)	9.00 (2.40, 50.65)	89.60 (43.10, 130.20)	**<0.001**
Leucocytes (10^9/L)	12.7 (9.5, 16.5)	12.5 (9.3, 16.1)	13.9 (11.1, 19.2)	0.064
Thrombocytes (10^9/L)	189 (145, 244)	187 (144, 241)	207 (155, 304)	0.268
Bilirubin (µmol/L)	9 (6, 15)	9 (6, 15)	12 (7, 17)	0.233
Albumin (g/L)	30 (25, 34)	30 (26, 34)	28 (22, 33)	0.084
Creatinine (µmol/L)	73 (58, 90)	69 (58, 82)	162 (136, 208)	**<0.001**
PO_2/_/FiO2	32.7 (23.7, 43.9)	33.1 (25.2, 44.6)	26.1 (18.5, 36.3)	**0.005**
Calprotectin (µg/mL)	4.17 ( 2.46, 7.00)	3.91 (2.36, 6.62)	5.06 (3.94, 9.92)	**0.011**
Thiols (µmol/L)	255.58 (161.63, 374.88)	275.57 (166.96, 395.21)	182.75 (75.78, 260.17)	**<0.001**
APACHE II	15 (11, 19)	14 (11, 18)	22 (18, 28.5)	**<0.001**
APACHE IV	48 (34, 65)	44 (33, 58)	89 (66, 104)	**<0.001**
<7-day Mortality (%)	17 (5.9)	11 (4.3)	6 (18.2)	**0.006**
<28-day Mortality (%)	30 (10.5)	20 (7.9)	10 (30.3)	**<0.001**

Data are presented as median (IQR) or proportions with corresponding percentages (%). *p*-values were calculated using a two-tailed Mann Whitney U test or chi-square test, while significant differences are indicated in bold. COPD, chronic obstructive pulmonary disease; CVD, cardiovascular disease; DM, diabetes mellitus; SIRS, systemic inflammatory response syndrome; SOFA, sequential organ failure assessment score; RRT, renal replacement therapy; MAP, mean arterial pressure; CRP, C-reactive protein; and APACHE, Acute Physiology And Chronic Health Evaluation.

**Table 2 antioxidants-09-01135-t002:** Plasma thiols are associated with AKI in the multivariable logistic regression model.

Factors	Univariate		Multivariate	
	Odds Ratio (95% CI)	*p*-Value	Odds Ratio (95% CI)	*p*-Value
Constant			−3.848 (−5.283 to –2.577)	<0.001
Plasma thiols (µmol/g)	−0.119 (−0.208 to –0.035)	0.007	−0.106 (−0.246 to –0.005)	0.047
APACHE IV	0.043 (0.030 to 0.056)	<0.001	0.039 (0.026 to 0.054)	<0.001
Diabetes mellitus	1.056 (0.352 to 1.735)	0.003	1.032 (0.122 to 1.924)	0.024
Admission via OR	−1.649 (−2.327 to –1.006)	<0.001		
CRP (mg/L)	0.006 (0.003 to 0.010)	<0.001		
Sepsis	1.510 (0.779 to2.239)	<0.001		

Plasma thiol levels, APACHE, and diabetes mellitus scores were associated with acute kidney injury (AKI). Other factors that entered the model were admission via the OR, operation room; CRP, C-reactive protein; and sepsis. Plasma thiol levels were adjusted for serum albumin. Model characteristics: chi-square = 77.003, degrees of freedom (df) = 4, N = 290, *p* < 0.001.

**Table 3 antioxidants-09-01135-t003:** Age and serum albumin are associated with plasma thiol levels in the multivariable linear regression model.

Factors	Univariate		Multivariate	
	B (95% CI)	*p*-Value	B (95% CI)	*p*-Value
Constant			6.158 (2.953 to 9.364)	<0.001
Age (in 10 years)	−1.065 (−1.401 to −0.728)	<0.001	−0.719 (−1.019 to −0.421)	<0.001
APACHE IV	−0.037 (−0.057 to −0.018)	<0.001		
CRP (mg/L)	−0.018 (−0.025 to −0.010)	<0.001	−0.008 (−0.015 to −0.001)	0.023
Serum albumin (g/L)	0.498 (0.426 to 0.567)	<0.001	0.470 (0.396 to0.544)	<0.001
Serum creatinine (µmol/L)	−0.007 (−0.010 to −0.001)	0.050	−0.009 (−0.015 to −0.003)	0.004
Serum NGAL (mg/mL)	−0.001 (−0.002 to −0.001)	0.215	−0.002 (<−0.001 to −0.003)	0.007
Infection	−2.364 (−3.790 to −0.937)	0.001		
Sepsis	−4.295 (−2.635 to −0.985)	0.002		
Diabetes Mellitus	−1.832 (−3.268 to −0.395)	0.013		

The thiols are transformed as the square root of thiols. Median thiol levels: 241.7 (146.9–347.3). Other factors that entered the model were calprotectin, gender, admission via OR, body mass index (BMI), CKD, CVD, DM, malignancy, SIRS, SOFA score, mechanical ventilation, heart rate, MAP, respiratory rate, body temperature, leucocytes, thrombocytes, and bilirubin. Model characteristics: R^2^ = 0.499, df = 5, N = 249, and *p* < 0.001. Calprotectin, gender, admission via OR, body mass index (BMI), CKD, CVD, SIRS score, mechanical ventilation, heart rate, respiratory rate, body temperature, leucocytes, thrombocytes, and bilirubin were excluded, because *p* > 0.200 in the univariate analysis. CKD, chronic kidney disease; CVD, cardiovascular disease; DM, diabetes mellitus; SIRS, systemic inflammatory response syndrome; SOFA, sequential organ failure assessment score; CPR, cardiopulmonary resuscitation; MAP, mean arterial pressure; CRP, C-reactive protein; NGAL, neutrophil gelatinase-associated lipocalin; and APACHE, Acute Physiology And Chronic Health Evaluation.

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
