# Peer review of "Acute Kidney Injury is Associated with Lowered Plasma-Free Thiol Levels"

_antioxidants, 2020, doi:10.3390/antiox9111135_

Round 1
Reviewer 1 Report
Paper present results of clinical study dealing with the critically ill patients with acute kidney injury with the aim to detect differentiating parameter(s) in other clinical conditions such as sepsis. The general design of the study was suitably chosen, the structure of the paper is reasonable and results seems sound which support the novelty and originality of the study. However, several questions or note shooul be clarified and explained.
- In the conlusion the expression "lower free thiol levels are reliably indicative of increased levels of oxidative stress" is not relevant and should be modified
- As authors noted, study was undertaken in the tertiary acdemic centre, where the coincidentical comorbidity of the patients cohort should interfere with the thiol level, reviewer suggest to reanalyze results within this note
- Ellman SH group determination is quite unspecific. Reliabililty of the study will be improved if additional analysis of oxidative stress such as total antioxidant capacity will be added, or determination of gluthatione (kidney failure might affect antioxidant vitamine level)
Author Response
To: Prof. Dr. Luciano Saso Antioxidants
Groningen, Oktober 29, 2020
Dear Dr. Luciano Saso,
We kindly thank you and the reviewers for your kind review of our manuscript. In agreement with the points raised by the reviewers we have now revised our manuscript, as described in the attached point-by-point response to the comments.
We used the provided suggestions and requested revisions to further improve the quality of our work, although some criticisms could be answered in our response to the reviewers and did not require editing of the manuscript. We have no requested revisions that we disagreed with.
We feel that the implementation of the issues raised by the reviewers allowed us to further improve the quality of our work and hope that our manuscript in its current form will qualify for publication in your journal.
Yours sincerely,
Lisanne Boekhoud, MD/PhD candidate
Reviewer 1: Paper present results of clinical study dealing with the critically ill patients with acute kidney injury with the aim to detect differentiating parameter(s) in other clinical conditions such as sepsis. The general design of the study was suitably chosen, the structure of the paper is reasonable and results seems sound which support the novelty and originality of the study. However, several questions or note should be clarified and explained. In the conclusion the expression "lower free thiol levels are reliably indicative of increased levels of oxidative stress" is not relevant and should be modified
Reply: We thank the reviewer for his/her kind review of our work. According to the reviewer’s suggestion, we have modified and rewritten the following sentence "As thiols are central components of the extracellular antioxidant network and key transducing elements in redox signaling, lower free thiol levels are reliably indicative of increased levels of oxidative stress" into “As thiols are central components of the extracellular antioxidant network and represent key transducing elements in redox signaling, they are indicative of oxidative stress” in the conclusion.
As authors noted, study was undertaken in the tertiary academic centre, where the coincidental comorbidity of the patients cohort should interfere with the thiol level, reviewer suggest to reanalyze results within this note.
Reply: Indeed, not only age and critical illness, but also (chronic) co-morbidity may have affected the levels of plasma free thiols. Therefore, we have carefully documented the co-morbidity of patients in our cohort (Table 1). To adjust for potential confounding by co-morbidity, we included the co-morbid diseases to adjust plasma free thiols in the multivariate regression (line no 109).
Ellman SH group determination is quite unspecific. Reliability of the study will be improved if additional analysis of oxidative stress such as total antioxidant capacity will be added, or determination of gluthatione (kidney failure might affect antioxidant vitamine level).
Reply: We thank the reviewer for raising this important comment. The reviewer is correct that plasma free thiol determination is quite unspecific in relation to oxidative stress-mediated diseases. However, it is not that unspecific with regard to the quantification of systemic oxidative stress [1,2,3]. Free thiols comprise the main biological targets of reactive species in the human body and govern multiple (protein) functions, enabling a variety of biological adaptations regulated by redox signaling pathways (e.g. modification of protein structure or activity, or modification of regulatory nodes and thereby affecting gene expression and gene regulation). The extracellular or systemic pool of free thiols may be viewed as a systemic redox buffer, as it serves as a communication conduit between the supply of nutritional precursors of reactive species and the full set of intracellular thiol targets.
Within the total antioxidant capacity, systemic free thiols lie at the center of the reactive species interactome (RSI) and they act as the major scavengers of reactive species, in addition to the relatively minor antioxidant contributions of, for example, bilirubin and uric acid. As they provide a robust monitoring tool for translational studies, they have been analyzed as integrative biomarker of systemic oxidative stress in a variety of oxidative-stress mediated conditions before, e.g. renal disease, cardiovascular disease, diabetes mellitus, and inflammatory bowel disease.
Other antioxidant compounds such as glutathione, homocysteine, or cysteine, comprise low-molecular-weight (LMW) thiols, and are of minor importance to the antioxidant capacity, as the greatest share is represented by protein free thiols (60-75%[4]). In addition, it is not always the case that they accurately represent the overall redox state or are part of a central redox hub, and instead reflect spill-over products origination from upstream inflammatory processes. Thus far, existing knowledge is insufficient to define the relative contributions of all these individual compounds and their specific roles in redox signaling pathways [1]. To elaborate on the central role of thiols as modulators of oxidative stress, we added the following sentence to the introduction: “Other antioxidant compounds such as glutathione, homocysteine, or cysteine, comprise low-molecular-weight (LMW) thiols, and are of minor importance to the antioxidant capacity, as the greatest share is represented by free thiols (60-75%) and thiols are the major determinants of the total antioxidant capacity [16]”.
Finally, as we agree with the reviewer that measuring additional redox molecules would always provide more granular insight into the observed associations, we attempted to do so by measuring total anti-oxidant capacity (TAC) in (a subset of) our study samples. However, we observed that the samples were in such a bad condition, presumably due to multiple freeze/thaw cycles, that TAC could not be quantified reliably.
Reviewer 2 Report
Lisanne Boekhoud and colleagues have investigated " Plasma Thiol Redox Status as Indicator of Acute Kidney Injury".
In the well-conducted work they could show that patients with sepsis and acute kidney damage had statistically significant lower plasma thiol levels. This observation is not entirely new. The authors list two studies (30,31) that demonstrated lowered thiol levels in both children and adults compared to the healthy population. Unfortunately, in the discussion the authors failed to distinguish their own results from these studies.
With regard to acute kidney damage, additional information is needed:
1. On the one hand, there is no stratification regarding the severity of renal damage. Statements on the necessity of renal replacement therapy are missing, but would be especially helpful for the following discussion (Thiol as prognostic marker). Of particular interest would be whether the patients were oliguric / anuric and thereofore a higher degree of tubular damage.
2. Further stratification (e.g. correlation to the SOFA score) would also be helpful regarding the correlation to sepsis.
It is conceivable that the renal damage per se is the cause of oxidative stress. However, the distinction from chronic renal function impairment is missing in the discussion. As an additional biomarker, the authors could/should determine and correlate osteopontin. In addition, other markers for oxidative stress should be measured and listed.
Whether antioxidants reduce the risk of AKI is still controversial. The data on disease is weak and currently only a hypothesis at best, so that this statement should be tempered.
Since the data in T&able of AKI shows strong fluctuations and overlapping (fig. 2 A and B) regarding plasma thiol levels / Serum Albumin, the discriminating effect regarding AKI for the individual would be (very) low. Accordingly, the last sentence ("may become an useful pathophysiological indicator of AKI") should be modified and mitigated.
Author Response
To: Prof. Dr. Luciano Saso Antioxidants
Groningen, Oktober 29, 2020
Dear Dr. Luciano Saso,
We kindly thank you and the reviewers for your kind review of our manuscript. In agreement with the points raised by the reviewers we have now revised our manuscript, as described in the attached point-by-point response to the comments.
We used the provided suggestions and requested revisions to further improve the quality of our work, although some criticisms could be answered in our response to the reviewers and did not require editing of the manuscript. We have no requested revisions that we disagreed with.
We feel that the implementation of the issues raised by the reviewers allowed us to further improve the quality of our work and hope that our manuscript in its current form will qualify for publication in your journal.
Yours sincerely,
Lisanne Boekhoud, MD/PhD candidate
Reviewer 2: Lisanne Boekhoud and colleagues have investigated " Plasma Thiol Redox Status as Indicator of Acute Kidney Injury". In the well-conducted work they could show that patients with sepsis and acute kidney damage had statistically significant lower plasma thiol levels. This observation is not entirely new. The authors list two studies (30,31) that demonstrated lowered thiol levels in both children and adults compared to the healthy population. Unfortunately, in the discussion the authors failed to distinguish their own results from these studies.
Reply: We thank the reviewer for his/her compliments and kind suggestion on this point. We respectfully disagree with the author that our findings are not novel, but do understand the potential confusion given the available literature on this topic. To clarify this and also better differentiate our results from current literature, we added the following sentence to the strengths and limitation part of the discussion: “To date, to the best of our knowledge, no other study unravelled the association between (albumin-adjusted) plasma free thiols and AKI in patients on the ICU, with and without sepsis. However, previous studies demonstrated lowered plasma free thiol levels in children with AKI [31] or adult patients with AKI [30], although not among adult patients on the ICU. Moreover, as thiols are influenced by serum albumin levels, which might be affected by critical illness or sepsis, we corrected the thiol levels for albumin, in contrast to previous studies.” Further, we added the following sentence in the discussion “Both the unadjusted thiol levels correlated with the albumin adjusted thiol levels in our study.” in line no 251.
With regard to acute kidney damage, additional information is needed:
1. On the one hand, there is no stratification regarding the severity of renal damage. Statements on the necessity of renal replacement therapy are missing, but would be especially helpful for the following discussion (Thiol as prognostic marker). Of particular interest would be whether the patients were oliguric / anuric and therefore a higher degree of tubular damage.
- Further stratification (e.g. correlation to the SOFA score) would also be helpful regarding the correlation to sepsis.
- It is conceivable that the renal damage per se is the cause of oxidative stress. However, the distinction from chronic renal function impairment is missing in the discussion. As an additional biomarker, the authors could/should determine and correlate osteopontin.
Reply:
- We agree with the reviewer that stratification according to AKI severity might reveal additional information about the role of thiols in the pathophysiology of AKI or could be used as biomarker to indicate the extent of AKI. However, in this cohort only 12 have severe AKI, as indicated by oliguria/anuria or need for renal replacement therapy. Although the numbers are very limited, we compared plasma free thiol levels of patients with severe AKI to patients with non-severe AKI and without AKI, which was not significantly different. To reveal this observation, we added the following sentence into the result section: “To assess whether plasma free thiol levels are also associated with the severity of AKI, we stratified patients into a group of severe AKI (oliguria, anuria or need of renal replacement therapy, n = 12), non-severe AKI (n = 21) and no AKI (n = 286) and compared plasma free thiol levels between groups. Levels of free thiols in plasma were not different between groups (p > 0.05), which might, however, be due to the relatively small group size after stratification.” in line no 131.
- Indeed, associating sepsis severity with the plasma free thiol level and AKI might provide additional information about the role of thiols in the pathophysiology of sepsis-AKI. In our cohort were 48 patients with sepsis; the SOFA score did not correlate with plasma free thiols levels and stratification to no sepsis, mild sepsis (SOFA < 8) and severe sepsis (SOFA sore >/= 8) did reveal a significant difference between groups (see graph below). Likely, the limited group size precludes identification of potential differences between groups. Given the likelihood of this type II error, we decided to not include these data in the current manuscript.
- The reviewer suggests to provide information about CKD and measure additional biomarkers of oxidative stress. First, all CKD patients were excluded from this study, which is now clarified in the method section as follows: “Patients with chronic kidney disease (CKD, defined by a previously known SCr >177 mmol/L) and patients on chronic RRT were excluded from the study, as were renal transplant recipients”. Next, all calculation were redone (see tables and figures).
In addition, the reviewer suggests to measure other markers for oxidative stress as well, such as osteopontin.
Reply: The reviewer is correct that plasma free thiol determination is quite unspecific in relation to oxidative stress-mediated diseases. However, it is not that unspecific with regard to the quantification of systemic oxidative stress [1, 2, 3]. Free thiols comprise the main biological targets of reactive species in the human body and govern multiple (protein) functions, enabling a variety of biological adaptations regulated by redox signaling pathways (e.g. modification of protein structure or activity, or modification of regulatory nodes and thereby affecting gene expression and gene regulation). The extracellular or systemic pool of free thiols may be viewed as a systemic redox buffer, as it serves as a communication conduit between the supply of nutritional precursors of reactive species and the full set of intracellular thiol targets.
Within the total antioxidant capacity, systemic free thiols lie at the center of the reactive species interactome (RSI) and they act as the major scavengers of reactive species, in addition to the relatively minor antioxidant contributions of, for example, bilirubin and uric acid. As they provide a robust monitoring tool for translational studies, they have been analyzed as integrative biomarker of systemic oxidative stress in a variety of oxidative-stress mediated conditions before, e.g. renal disease, cardiovascular disease, diabetes mellitus, and inflammatory bowel disease.
Other antioxidant compounds such as glutathione, homocysteine, or cysteine, comprise low-molecular-weight (LMW) thiols, and are of minor importance to the antioxidant capacity, as the greatest share is represented by protein free thiols (60-75% [4]). In addition, it is not always the case that they accurately represent the overall redox state or are part of a central redox hub, and instead reflect spill-over products origination from upstream inflammatory processes. Thus far, existing knowledge is insufficient to define the relative contributions of all these individual compounds and their specific roles in redox signaling pathways [1]. To elaborate on the central role of thiols as modulators of oxidative stress, we added the following sentence to the introduction: “
Other antioxidant compounds such as glutathione, homocysteine, or cysteine, comprise low-molecular-weight (LMW) thiols, and are of minor importance to the antioxidant capacity, as the greatest share is represented by free thiols (60-75%) and thiols are the major determinants of the total antioxidant capacity [16]”.
Finally, as we agree with the reviewer that measuring additional redox molecules would always provide more granular insight into the observed associations, we attempted to do so by measuring total anti-oxidant capacity (TAC) in (a subset of) our study samples. However, we observed that the samples were in such a bad condition, presumably due to multiple freeze/thaw cycles, that TAC could not be quantified reliably.
Whether antioxidants reduce the risk of AKI is still controversial. The data on disease is weak and currently only a hypothesis at best, so that this statement should be tempered.
Reply: According to the reviewer’s suggestion, we have rewritten the following sentence in line no 279: “Since plasma free thiols were decreased in AKI, it might also identify patients at risk for the development of AKI.” into “The lowered plasma free thiol levels in patients with AKI might either indicate scavenging of anti-oxidants due to the increased oxidative stress, or, vice versa, suggest a protective effect of increased plasma free thiol levels against AKI. Yet, the current data do not allow to draw a firm conclusion about the precise role in the pathophysiology of AKI.”
Since the data in Table of AKI shows strong fluctuations and overlapping (fig. 2 A and B) regarding plasma thiol levels / serum albumin, the discriminating effect regarding AKI for the individual would be (very) low. Accordingly, the last sentence ("may become an useful pathophysiological indicator of AKI") should be modified and mitigated.
Reply: We agree with the reviewer, therefore we modified the sentence "May become a useful pathophysiological indicator of AKI” into “The pathophysiology of AKI in critical illness, whether due to sepsis or other conditions, remains complex and incompletely understood. Although we reveal a correlation between lowered plasma free thiol levels and AKI, which confirms the hypothesis that oxidative stress plays a key role in the pathophysiology of AKI on the ICU. Although, at least for some individuals in our cohort, the lowered plasma thiol level might well contribute to the development of AKI, the discriminative effect of plasma free thiol levels is limited, illustrating the complex pathophysiology in this relatively heterogeneous population.”
Reviewer 3 Report
General comments:
The topic is important, however, the data presented is far preliminary for publication. The modest methodological approach gives rise to only a limited data set. Additionally, many concern emerges about the design of the study, the accuracy of the methodology used, and the interpretation of the data which are outlined in point form below.
- Patients having AKI need not suffer from Sepsis. Pathophysiological conditions and clinical parameters will completely differ in the case of patients suffering only with AKI in contrast to patients developing AKI due to sepsis. As per the current 2016 SCCM/ESICM evaluation of criteria for identifying septic patients, including the Logistic Organ Dysfunction System (LODS) and Sequential Organ Failure Assessment (SOFA) scoring there are other additional severities of vital organs which is not clearly stated in the manuscript.
- In the line no 104 – SOFA score is wrongly stated instead of the SIRS criteria, namely tachycardia (heart rate >90 beats/min), tachypnea (respiratory rate >20 breaths/min), fever or hypothermia (temperature >38 or <36 °C), and leukocytosis, leukopenia, or bandemia (white blood cells >1,200/mm3, <4,000/mm3 or bandemia ≥10%) are defined as Sepsis-1, which was established at a 1991 consensus conference. Patients who met two or more of these criteria fulfilled the definition of SIRS, and Sepsis-1.
- The below mentioned clinical parameters are missing in the manuscript when SOFA score is considered in the septic patients developing AKI (Acute Kidney Injury)
Ref: doi: 10.21037/jtd.2017.03.125
Author Response
To: Prof. Dr. Luciano Saso Antioxidants
Groningen, Oktober 29, 2020
Dear Dr. Luciano Saso,
We kindly thank you and the reviewers for your kind review of our manuscript. In agreement with the points raised by the reviewers we have now revised our manuscript, as described in the attached point-by-point response to the comments.
We used the provided suggestions and requested revisions to further improve the quality of our work, although some criticisms could be answered in our response to the reviewers and did not require editing of the manuscript. We have no requested revisions that we disagreed with.
We feel that the implementation of the issues raised by the reviewers allowed us to further improve the quality of our work and hope that our manuscript in its current form will qualify for publication in your journal.
Yours sincerely,
Lisanne Boekhoud, MD/PhD candidate
Reviewer 3: The topic is important, however, the data presented is far preliminary for publication. The modest methodological approach gives rise to only a limited data set. Additionally, many concern emerges about the design of the study, the accuracy of the methodology used, and the interpretation of the data which are outlined in point form below.
Reply: We thank the reviewer for his/her kind review of our work. Although we do agree that the data is preliminary, we think this is important to publish. However, our results confirm the hypothesis that lowered plasma free thiols are associated with AKI, for which we used a relevant population among which oxidative stress and AKI are very common, and employed the current standard in measuring plasma free thiol levels. We agree that the current results are too preliminary to draw firm conclusions about the role of thiols in the pathophysiology of AKI and their potential as (amenable) biomarker, which should (and will) be topic of follow-up studies.
Patients having AKI need not suffer from Sepsis. Pathophysiological conditions and clinical parameters will completely differ in the case of patients suffering only with AKI in contrast to patients developing AKI due to sepsis. As per the current 2016 SCCM/ESICM evaluation of criteria for identifying septic patients, including the Logistic Organ Dysfunction System (LODS) and Sequential Organ Failure Assessment (SOFA) scoring there are other additional severities of vital organs which is not clearly stated in the manuscript.
Reply: We agree that sepsis and AKI are two different entities and apologizes for the misinterpretation, which is probably due to our statements describing that AKI is most commonly due to sepsis, and vice versa, sepsis is commonly associated with AKI. To clarify this, we included the following sentence in the results section: “In total 44.8% of the patients with AKI in our study also had sepsis.” in line no 104. Although the pathophysiology of sepsis-AKI might be different from other types of AKI, we here studied a population with critical illness on the ICU, where oxidative stress and hypoperfusion are considered key players in the pathophysiology of AKI. To better differentiate the role of thiols in sepsis and their effect on AKI, we have decided to physically separate these paragraphs in the discussion section. Finally, the reviewer’s suggestion was to add the LODS, we did not do this, because we miss the prothrombin time and urea. We provided more detailed information about the SOFA scores in this cohort (Table 1) and mentioned that sepsis was based using the current Sepsis-3 criteria as follows in the results section: “The diagnosis of sepsis was based on a SOFA score of two or more, which is based on PaO2/FiO2 ratio, thrombocyte count, bilirubin level, mean arterial pressure (MAP) and need for vasopressors, Glasgow coma scale and kidney function (i.e. based on serum creatinine and diuresis).”
In the line no 104 – SOFA score is wrongly stated instead of the SIRS criteria, namely tachycardia (heart rate >90 beats/min), tachypnea (respiratory rate >20 breaths/min), fever or hypothermia (temperature >38 or <36 °C), and leukocytosis, leukopenia, or bandemia (white blood cells >1,200/mm3, <4,000/mm3 or bandemia ≥10%) are defined as Sepsis-1, which was established at a 1991 consensus conference. Patients who met two or more of these criteria fulfilled the definition of SIRS, and Sepsis-1.
Reply: It is unclear where the reviewer is referring to, as we do not mention the SIRS-criteria in our manuscript, other than to indicate the incidence of sepsis according to Sepsis-1 or Sepsis-2 criteria in table 1. We agree with the reviewer that the current Sepsis-3 criteria define another population than the older sepsis criteria. Therefore, to increase the clarity of the different definitions we added: “The diagnosis of sepsis was based on a SOFA score of two or more, which is based on PaO2/FiO2 ratio, thrombocyte count, bilirubin level, mean arterial pressure (MAP) and need for vasopressors, Glasgow coma scale and kidney function (i.e. based on serum creatinine and diuresis).” in line no 105 in the results section.
The below mentioned clinical parameters are missing in the manuscript when SOFA score is considered in the septic patients developing AKI (Acute Kidney Injury).
Reply: We agree with the reviewer. The missing parameters of the SOFA score were Glasgow coma scale and the PaO2/FiO2, which have now been added to the baseline table.
Reviewer 4 Report
The present study shows the results of the association between albumin-adjusted plasma free thiol levels and sepsis-associated AKI, as potential biomarkers for oxidative stress.
The paper is well-written, the results are presented in a concise form.
However, it is not very clear when was the exact time during patient admission that plasma free thiol levels were determined. Was it from plasma samples collected at day 1 or every day? The same for the other laboratory values (CRP, leukocytes, thrombocytes, bilirubin) and also calprotectin and NGAL. For acute conditions, correlations should be done between parameters determined from samples collected at approximately the same time.
It would be interesting to show a dynamic evolution of plasma free thiols in patients according to their evolution (favorable vs. unfavorable), to see if plasma free thiols have a prognostic value during hospital admission.
Author Response
To: Prof. Dr. Luciano Saso Antioxidants
Groningen, Oktober 29, 2020
Dear Dr. Luciano Saso,
We kindly thank you and the reviewers for your kind review of our manuscript. In agreement with the points raised by the reviewers we have now revised our manuscript, as described in the attached point-by-point response to the comments.
We used the provided suggestions and requested revisions to further improve the quality of our work, although some criticisms could be answered in our response to the reviewers and did not require editing of the manuscript. We have no requested revisions that we disagreed with.
We feel that the implementation of the issues raised by the reviewers allowed us to further improve the quality of our work and hope that our manuscript in its current form will qualify for publication in your journal.
Yours sincerely,
Lisanne Boekhoud, MD/PhD candidate
Reviewer 4: The present study shows the results of the association between albumin-adjusted plasma free thiol levels and sepsis-associated AKI, as potential biomarkers for oxidative stress. The paper is well-written, the results are presented in a concise form. However, it is not very clear when was the exact time during patient admission that plasma free thiol levels were determined. Was it from plasma samples collected at day 1 or every day? The same for the other laboratory values (CRP, leukocytes, thrombocytes, bilirubin) and also calprotectin and NGAL. For acute conditions, correlations should be done between parameters determined from samples collected at approximately the same time.
Reply: We thank this reviewer for his/her kind feedback. We agree with the reviewer that the moment of thiol measurement is not entirely clear from the paper. Hence, we added “Thiols were measured in samples derived on the first day of ICU admission, while creatinine was measured on the first and third day of ICU admission to determine AKI progression.” in the method section in line no 335.
It would be interesting to show a dynamic evolution of plasma free thiols in patients according to their evolution (favorable vs. unfavorable), to see if plasma free thiols have a prognostic value during hospital admission.
Reply: We fully agree with the reviewer that exploring the kinetics of plasma free thiols during ICU stay might reveal additional information about the potential prognostic role of thiols. However, in the current project we had only samples from the first day of ICU admission available to measure plasma free thiols. Moreover, measuring plasma free thiols at multiple time-points are beyond the scope of the current project, in which we aimed to reveal – as proof-of-principle – that lower plasma free thiol levels are associated with AKI. To determine whether the change in plasma free thiols over time might be of prognostic relevance in predicting AKI, we are currently collecting samples in a clinical study to follow-up the current project.
Reference:
- Cortese-Krott MM, Koning A, Kuhnle GGC, Nagy P, Bianco CL, Pasch A, Wink DA, Fukuto JM, Jackson AA, van Goor H, Olson KR, Feelisch M. The Reactive Species Interactome: Evolutionary Emergence, Biological Significance, and Opportunities for Redox Metabolomics and Personalized Medicine. Antioxid Redox Signal. 2017 Oct 1;27(10):684-712. doi: 10.1089/ars.2017.7083. Epub 2017 Jun 6. PMID: 28398072; PMCID: PMC5576088.
- Bourgonje AR, Feelisch M, Faber KN, Pasch A, Dijkstra G, van Goor H. Oxidative Stress and Redox-Modulating Therapeutics in Inflammatory Bowel Disease. Trends Mol Med. 2020 Jun 30:S1471-4914(20)30157-X. doi: 10.1016/j.molmed.2020.06.006. Epub ahead of print. PMID: 32620502.
- Rudyk O, Eaton P. Biochemical methods for monitoring protein thiol redox states in biological systems. Redox Biol. 2014 Jun 13;2:803-13. doi: 10.1016/j.redox.2014.06.005. PMID: 25009782; PMCID: PMC4085346.
- Turell L, Radi R, Alvarez B. The thiol pool in human plasma: The central contribution of albumin to redox processes. Free Radic Biol Med 2013, 65, 244-253. doi: 10.1016/j.freeradbiomed.2013.05.050
Round 2
Reviewer 2 Report
Thank you for the opportunity to critically reflect on the work about the associatio of Thiol Redox Status and AKI. The authors have incorporated the comments of the reviewers and thus substantial improved the manuscript.
In my opinion, there are still two major points of criticism that need to be modified:
1) The title of the paper ("Plasma Thiol Redox Status as Indicator of Acute Kidney Injury") is misleading as it clearly exaggerates the value of the diagnosis of an AKI. According to the results presented, "Acute kidney injury is associated with lowered plasma free thiol levels" would be much more appropriate.
2) All patients in Table 1 (Baseline characteristics) have a Glagow Coma Scale score of 15! This is not only a very surprising finding in itself, as it would obviously mean that all patients were completely free of neurological impairment. In view of the considerable septic population (45%), this finding is simply not reliable and should be checked again. Furthermore, it is not clear how a statistical significance of 0.06 can be achieved with three identical values.
Author Response
To: Prof. Dr. Luciano Saso Antioxidants
Groningen, November 5, 2020
Dear Dr. Luciano Saso,
We kindly thank you and the reviewers for your review of our manuscript and agreement to publish pending minor improvements. In agreement with the points raised by the reviewers we have now revised our manuscript, as described in the attached point-by-point response to the comments.
We used the provided suggestions and requested revisions to further improve the quality of our work, although some criticisms could be answered in our response to the reviewers and did not require editing of the manuscript. We have no requested revisions that we disagreed with.
We feel that the implementation of the issues raised by the reviewers allowed us to further improve the quality of our work and hope that our manuscript in its current form will now qualify for publication in your journal.
Yours sincerely,
Lisanne Boekhoud, MD/PhD candidate
Reviewer 2: Thank you for the opportunity to critically reflect on the work about the associatio of Thiol Redox Status and AKI. The authors have incorporated the comments of the reviewers and thus substantial improved the manuscript.
In my opinion, there are still two major points of criticism that need to be modified:
- The title of the paper ("Plasma Thiol Redox Status as Indicator of Acute Kidney Injury") is misleading as it clearly exaggerates the value of the diagnosis of an AKI. According to the results presented, "Acute kidney injury is associated with lowered plasma free thiol levels" would be much more appropriate.
Reply: We thank the reviewer for this suggestion. We changed the title from “Plasma Thiol Redox Status as Indicator of Acute Kidney Injury" to “Acute kidney injury is associated with lowered plasma free thiol levels” as the reviewer suggested.
- All patients in Table 1 (Baseline characteristics) have a Glagow Coma Scale score of 15! This is not only a very surprising finding in itself, as it would obviously mean that all patients were completely free of neurological impairment. In view of the considerable septic population (45%), this finding is simply not reliable and should be checked again. Furthermore, it is not clear how a statistical significance of 0.06 can be achieved with three identical values.
Reply: The reviewer is correc that the data suggests that all patients had normal consciousness as the median is 15. However, as shown by the IQR of 13-15 we have included patients with a GCS < 15. To improve presentation of the variation in GCS, we decided to present the incidence per GCS score in our data set as follows:
|
Characteristics |
Total (n=286) |
No AKI at admission (n=253 [88.4%]) |
AKI at admission (n=33 [11.6%]) |
p-value |
|
Glasgow coma scale |
|
|
|
0.013 |
|
15 |
229 (80.9) |
209 (83.3) |
20 (62.5) |
|
|
13 – 14 |
14 (4.9) |
10 (4.0) |
4 (12.5) |
|
|
≤12 |
40 (14.1) |
32 (12.7) |
8 (25.0) |
|
Author Response
To: Prof. Dr. Luciano Saso Antioxidants
Groningen, November 5, 2020
Dear Dr. Luciano Saso,
We kindly thank you and the reviewers for your review of our manuscript and agreement to publish pending minor improvements. In agreement with the points raised by the reviewers we have now revised our manuscript, as described in the attached point-by-point response to the comments.
We used the provided suggestions and requested revisions to further improve the quality of our work, although some criticisms could be answered in our response to the reviewers and did not require editing of the manuscript. We have no requested revisions that we disagreed with.
We feel that the implementation of the issues raised by the reviewers allowed us to further improve the quality of our work and hope that our manuscript in its current form will now qualify for publication in your journal.
Yours sincerely,
Lisanne Boekhoud, MD/PhD candidate
Reviewer 3: No comment of suggestions for authors were filled in, however as suggested we rewrote the following in the methods.
Reply: We thank the reviewer for this suggestions. To further improve readability of the Materials and Methods section, we critically reviewed and rewrote this section. In addition to language editing, we have changed the following content:
- The number of patients of CKD and RRT is added. Thereby, we have now included the following sentence in the methods section: “Patients with chronic kidney disease (CKD, defined by a previously known SCr >177 mmol/L) and patients on chronic RRT were excluded from the study, as were renal transplant recipients (n=15).”
- We changed the following sentence “plasma free thiol concentrations were measured as previously described by Ellman et al., with minor modifications ” to “plasma free thiol concentrations were measured as previously described by Ellman et al., with minor modifications”
- The following statement “Serum calprotectin levels were quantified using MRP8/14 ELISA kit (BÜHLMANN, Schönenbuch, Switzerland) using the DS2 ELISA robot (DS2, Dynex, Chantilly, USA) according to manufacturer’s instructions to determine the neutrophil activation” is modified to “Serum calprotectin levels were quantified using MRP8/14 ELISA kit (BÜHLMANN, Schönenbuch, Switzerland) using the DS2 ELISA robot (DS2, Dynex, Chantilly, USA) according to manufacturer’s instructions to determine the neutrophil activation, however they are also released though monocytes, macrophages and squamous epithelial cells.”
Reviewer 4 Report
The authors have adequately answered to all issues raised. Modifications to the manuscript are satisfactory.
Author Response
To: Prof. Dr. Luciano Saso Antioxidants
Groningen, November 5, 2020
Dear Dr. Luciano Saso,
We kindly thank you and the reviewers for your review of our manuscript and agreement to publish pending minor improvements. In agreement with the points raised by the reviewers we have now revised our manuscript, as described in the attached point-by-point response to the comments.
We used the provided suggestions and requested revisions to further improve the quality of our work, although some criticisms could be answered in our response to the reviewers and did not require editing of the manuscript. We have no requested revisions that we disagreed with.
We feel that the implementation of the issues raised by the reviewers allowed us to further improve the quality of our work and hope that our manuscript in its current form will now qualify for publication in your journal.
Yours sincerely,
Lisanne Boekhoud, MD/PhD candidate
Reviewer 4: The authors have adequately answered to all issues raised. Modifications to the manuscript are satisfactory.
Reply: Thank you for this compliment, we are pleased that the modifications were satisfactory.